# Neural activity related to volitional regulation of cortical excitability

Kathy Ruddy[1,2], Joshua Balsters[1,3], Dante Mantini[1,4], Quanying Liu[1,4], Pegah Kassraian-Fard[1], Nadja Enz[1], Ernest Mihelj[1], Bankim Subhash Chander[5], Surjo R Soekadar[5,6], Nicole Wenderoth[1]*

[1]Neural Control of Movement Lab, ETH Zürich, Zürich, Switzerland; [2]Institute of Neuroscience, Trinity College Dublin, Dublin, Ireland; [3]Department of Psychology, Royal Holloway University of London, London, United Kingdom; [4]Movement Control and Neuroplasticity Research Group, KU Leuven, Leuven, Belgium; [5]Applied Neurotechnology Laboratory, University of Tübingen, Tübingen, Germany; [6]Clinical Neurotechnology Laboratory, Neuroscience Research Center (NWFZ), Department of Psychiatry and Psychotherapy, Charité – University Medicine Berlin, Berlin, Germany

**Abstract** To date there exists no reliable method to non-invasively upregulate or downregulate the state of the resting human motor system over a large dynamic range. Here we show that an operant conditioning paradigm which provides neurofeedback of the size of motor evoked potentials (MEPs) in response to transcranial magnetic stimulation (TMS), enables participants to self-modulate their own brain state. Following training, participants were able to robustly increase (by 83.8%) and decrease (by 30.6%) their MEP amplitudes. This volitional up-versus down-regulation of corticomotor excitability caused an increase of late-cortical disinhibition (LCD), a TMS derived read-out of presynaptic $GABA_B$ disinhibition, which was accompanied by an increase of gamma and a decrease of alpha oscillations in the trained hemisphere. This approach paves the way for future investigations into how altered brain state influences motor neurophysiology and recovery of function in a neurorehabilitation context.

DOI: https://doi.org/10.7554/eLife.40843.001

*For correspondence:
nicole.wenderoth@hest.ethz.ch

**Competing interests:** The authors declare that no competing interests exist.

## Introduction

Rhythmic oscillatory brain activity at rest is associated with high versus low neuronal responsiveness, or 'excitability' of a region (*Jensen and Mazaheri, 2010*; *Jensen et al., 2011*). Measuring these momentary fluctuations of neural activity via electro- or magnetoencephalography (EEG/MEG) over human primary motor cortex (M1), it has been demonstrated that frequency, amplitude and phase of the ongoing oscillation cycle systematically modulate responses evoked by transcranial magnetic stimulation (TMS) (*Zarkowski et al., 2006*; *Sauseng et al., 2009*; *Schaworonkow et al., 2018*; *Kelly et al., 2009*; *Mazaheri et al., 2009*; *Schubert et al., 2006*). In particular, it has been shown that corticomotor excitability is significantly higher when the power (amplitude) of sensorimotor rhythms in the alpha band (8–14 Hz, also called the 'mu'-rhythm), or beta band (15–30 Hz) are low, or when M1 is stimulated during the trough of the oscillatory cycle of these rhythms (*Zaehle et al., 2010*). This concept has inspired neurofeedback interventions whereby, for example, stroke patients learn to volitionally desynchronize sensorimotor rhythms with the goal of bringing the sensorimotor system into a more excitable state as a precursor for enhanced neural plasticity and accelerated recovery (*Buch et al., 2008*; *Caria et al., 2011*; *Soekadar et al., 2015a*).

Previous research has focussed on interactions between corticomotor excitability and cortical dynamics at rest, but much less is known about whether it is possible to *voluntarily* control the

excitability of sensorimotor circuits while keeping motor output and sensory feedback constant. In the case of stroke rehabilitation, this mechanism may become particularly relevant as patients are unable to move or receive sensory feedback from the paretic limb. Therefore, interventions that optimally harness the residual ability to voluntarily and endogenously activate relevant brain circuits in the days and weeks after the incident, may provide the crucial innervation necessary to promote re-wiring for functional recovery (*Maulden et al., 2005*).

It is well known that primates (*Fetz, 2013*; *Engelhard et al., 2013*), and humans (*Buch et al., 2008*; *Mellinger et al., 2007*; *Soekadar et al., 2015b*; *Sitaram et al., 2017*; *Thompson et al., 2009*) can gain volitional control of neural activity by receiving neurofeedback via a brain-computer interface (BCI). Here, we used a BCI-neurofeedback approach as an effective method for training participants to both volitionally upregulate and downregulate corticomotor excitability as reflected by the size of TMS-evoked motor potentials (MEPs), with the aim to modulate their amplitudes over a much larger dynamic range than observed during rest. Using this approach enabled us to investigate the neural mechanisms that underlie volitional up- versus down-regulation of corticospinal excitability in the motor system and the associated oscillatory signatures. By modulating one neural marker, motor evoked potential amplitude, while measuring independent modalities using EEG, or paired-pulse TMS, this approach allows us to causally relate voluntary rather than incidental changes of corticomotor excitability to cortical dynamics.

To achieve this goal we developed a BCI by stimulating the cortex with TMS and providing neurofeedback of MEP amplitudes (*Figure 1*). The feedback was designed such that participants were rewarded for larger than average MEPs in one condition, and smaller than average in another condition.

In a within-subject cross-over design, participants performed four training sessions with TMS-neurofeedback of MEP amplitudes in order to learn how to up- and downregulate their corticomotor excitability (two 'UP' sessions, two 'DOWN' sessions, order counterbalanced across participants). After the training we characterised the neural underpinnings of these two distinct activity states in detail by conducting a series of multimodal experiments using EEG and paired pulse TMS to profile the associated oscillatory and neurophysiological processes. As it has been proposed that dynamic modulation of neuronal activity is realized via synchronization of high frequency rhythms (*Fries, 2005*) which are tightly coupled to desynchronizing sensorimotor rhythms (*Grosse-Wentrup et al., 2011*), we hypothesised that Gamma synchronisation (31–80 Hz) and alpha (8–13 Hz)/beta (14–30 Hz) desynchronization play a critical role in actively determining the state of the motor cortex. Specifically, we hypothesised that following UP neurofeedback training, MEP amplitudes would be increased, and this volitional upregulation would be associated with increased gamma synchronisation and alpha/beta desynchronization. Concurrent reductions in TMS-derived measures of inhibition were also hypothesised during upregulation. The reverse pattern was predicted following DOWN neurofeedback training.

## Results

### Bidirectional changes in corticospinal excitability were observed in the MEP neurofeedback group but not in a control group

We first tested whether participants could learn to volitionally increase or decrease (bidirectional) corticomotor excitability when using a motor imagery strategy shaped by neurofeedback of MEP amplitudes. Across two training sessions, we found that MEP amplitudes increased during UP training (*Figure 2A*, orange symbols) and decreased during DOWN training (*Figure 2A*, blue symbols, also see *Figure 2—figure supplements 2*, *3* and *4*) relative to the baseline measurement (BS), revealing a significant dissociation over time (*neurofeedback type* x *block number* interaction during training session 1 [$F_{(4,115.9)} = 3.87$, p = 0.006], session 2 [$F_{(4,125.0)} = 3.7$, p = 0.007] and EEG session [$F_{(2,70)} = 6.9$, p = 0.002], *F tests following mixed effects models*, n = 15; see *Figure 2—figure supplement 1* for additional analyses). Since MEP amplitudes are a compound measure of excitability influenced by multiple neural elements (*Carson et al., 2016*), including background muscle activity (*Hess et al., 1986*; *Devanne et al., 1997*), we repeated this analysis using the root mean squared (rms) background muscle activation (EMG) recorded in the 100 ms prior to each TMS pulse. Importantly, this control analysis revealed no such interactions on any of the sessions, suggesting that the

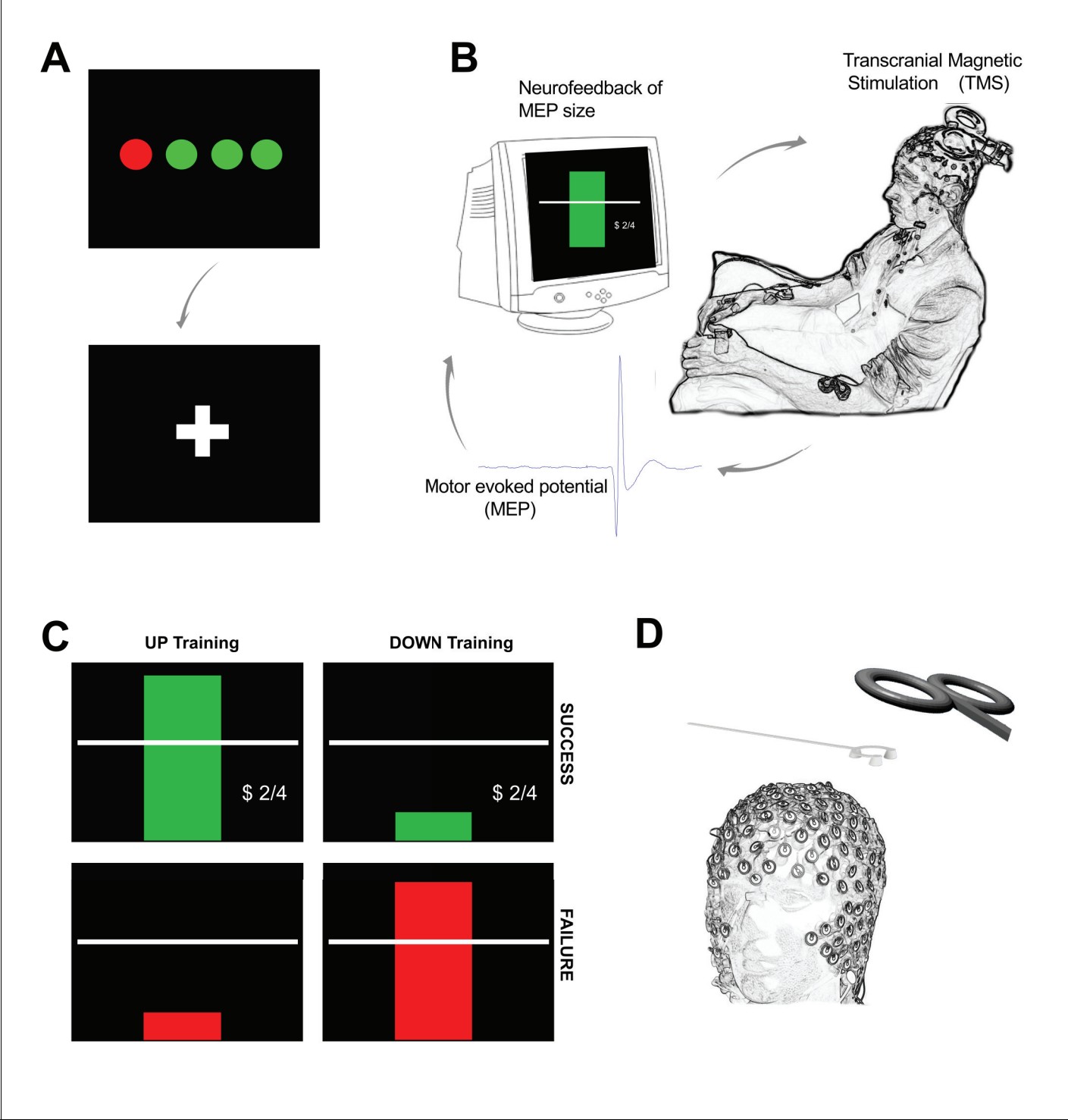

**Figure 1.** Outline of experimental setup. Each trial of neurofeedback training commenced with a display of four circles (A) each representing the background EMG in one of the recorded hand muscles (right FDI, ADM and OP, and left FDI). The circles were red if the root mean squared (rms) EMG at rest was greater than seven microvolts. It was essential that all four circles were green for at least 500 ms before the trial could proceed. When this condition was met a fixation cross appeared for a random period (in order to prevent anticipation of the TMS pulse). During the fixation period, it was still essential to keep the background EMG below seven microvolts in order for a TMS pulse to be delivered. (B) The peak-peak amplitude of the motor evoked potential (MEP) evoked by the TMS was calculated in real-time and displayed immediately to the participant on screen in the form of a rectangular bar. (C) Different feedback for UP training and DOWN training. In the UP training if the MEP was greater than the baseline mean, the rectangle was green, with a green tick, a dollar sign to indicate a small financial reward, a display of the current score, and a positive encouraging

*Figure 1 continued on next page*

*Figure 1 continued*

sound bite was heard. If the MEP did not meet the criterion amplitude, the bar was red, there was no dollar sign, and a negative sound bite was heard. (**D**) A custom 3D printed 'coil spacer' device was used to prevent direct contact of the TMS coil on the EEG electrodes and allow the pre-TMS EEG period to be recorded artefact free.

DOI: https://doi.org/10.7554/eLife.40843.002

The following figure supplement is available for figure 1:

**Figure supplement 1.** Power analysis to compute sample size.

DOI: https://doi.org/10.7554/eLife.40843.003

observed modulation was not driven by changes in peripheral activity of the target muscle, nor any of the additional three control muscles (OP, ADM, left FDI) (all p > 0.18, see *Supplementary file 1* Table 2, in *Supplementary file 1*). Further analyses of background EMG revealed no systematic variation in the number of rejected/retained MEP trials over the course of training, or by feedback type (See *Supplementary file 1* Table 3 and 4).

In order to isolate the effect of the neurofeedback, we included a control group who undertook the same protocol, using the same mental imagery strategies, but with feedback that was not contingent on the MEP amplitudes. This group exhibited no systematic changes of corticomotor excitability across training (*Figure 2B* and *Figure 2—figure supplements 2* and *3*) and mixed effects models

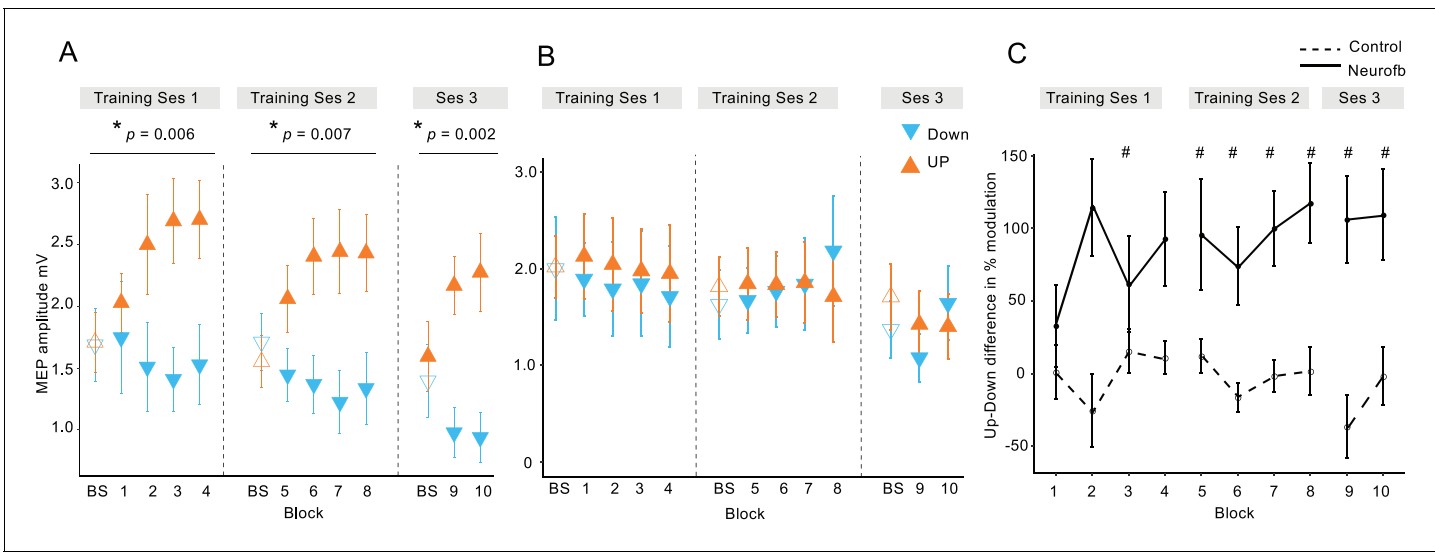

**Figure 2.** MEP amplitudes during neurofeedback. Panel (**A**) depicts MEP amplitude in millivolts during the two types of MEP neurofeedback. UP training is shown in orange and DOWN training in blue, across all 10 training blocks. Unfilled triangles labelled 'BS' indicate the baseline measurement block that occurred at the beginning of that particular session, prior to any neurofeedback. Dotted vertical lines indicate the separation of the blocks into different 'sessions', which occurred on separate days. Panel (**B**) shows the same data for the control group who received no veridical neurofeedback. Panel (**C**) shows the UP-DOWN difference (in the normalised % change from baseline data) for each block in the experimental group and the control group. Higher values represent greater deviations between the UP and DOWN data points and therefore more modulation of MEP amplitude. Thus, these values are significantly higher in the experimental group than in the control group. # symbols indicate blocks in which the Cohen's *d* effect size for the difference between the experimental and control group was large-very large (> 0.8). All data are shown as mean ± SEM.

DOI: https://doi.org/10.7554/eLife.40843.004

The following figure supplements are available for figure 2:

**Figure supplement 1.** Percentage change in MEP amplitude.

DOI: https://doi.org/10.7554/eLife.40843.005

**Figure supplement 2.** MEP amplitudes during neurofeedback for all participants.

DOI: https://doi.org/10.7554/eLife.40843.006

**Figure supplement 3.** Individual representative subjects' performance across trials.

DOI: https://doi.org/10.7554/eLife.40843.007

**Figure supplement 4.** Scatterplots showing individual MEP data trials across learning.

DOI: https://doi.org/10.7554/eLife.40843.008

revealed no significant *neurofeedback type* x *block number* interactions on any of the separate testing sessions in the control group (all p > 0.06, note that statistics approached significance for the second session because MEPs were randomly higher in the DOWN than in the UP condition; see *Figure 2—figure supplement 1* for further details). Additionally, there were no significant differences in background EMG (All p > 0.09, *Supplementary file 1* Table 2). Next we compared the performance of the experimental and the control group, by normalizing MEP amplitudes to baseline (% change) and calculating the difference between UP and DOWN (*Figure 2C*). The differences were substantial in the experimental group, who exhibited on average MEP amplitudes twice as large during UP than during DOWN, and differed significantly from the control group where systematic differences were virtually absent (effect of 'Group' [$F_{(1,25.6)}$ = 13.32, p = 0.001], *F tests following mixed effects*, n = 28). The effect sizes (Cohen's *d*) of the between-group differences were small for the first two blocks (< 0.5), but consistently increased during training (*d* = 1.27 for block 8), and remained high in the two blocks of the EEG session (*d* > 0.97). As the control group were executing the same mental imagery strategies as the experimental group, this comparison demonstrates that vertical TMS neurofeedback was essential for gaining volitional control over corticomotor excitability.

## Neurofeedback training effects are retained for at least 6 months

In a follow-up investigation approximately 6 months following initial neurofeedback training, we showed for a subset of the participants (n = 11) that they had retained the ability to upregulate and downregulate their MEP amplitude with neurofeedback (*Figure 3*; significant effect of neurofeedback type (UP vs DOWN) in a retention block carried out with no top-up training (F(1,10)=6.64, p = 0.028). Again, this was in the absence of any modulation in background muscle activation (*Supplementary file 1* Table 5). Measurements of resting MEP amplitude taken 5 and 10 min following the retention block indicated no after-effects (all p > 0.2) indicating that subjects could acutely control corticomotor excitability without long-lasting after-effects. Having verified that the ability to modulate brain states had been robustly retained, we next tested whether participants could sustain this performance even when feedback was removed. Performing a feedback-free block, we found that MEP amplitude was significantly larger in the UP versus DOWN condition (F(1,10) = 12.32, p = 0.006), indicating that when participants have reached a sufficient level of training they have optimised their mental strategies and no longer require continuous feedback.

In order to measure whether the feedback-induced changes in corticospinal excitability were specific to the hemisphere targeted by the neurofeedback, we used two TMS coils simultaneously and performed one block of 40 trials, where half of the TMS pules were applied to the left hemisphere (i.e. the usual feedback hemisphere), and the other half to the right (i.e. the opposite hemisphere). We found that the same pattern of upregulation and downregulation of MEP amplitudes was observed in the opposite hemisphere, an effect that approached significance (F(1,20) = 4.032, p = 0.07) but was much smaller than in the neurofeedback hemisphere particularly for the UP condition (UP *d* = 1.01 for neurofeedback hemi, *d* = 0.27 for opp. hemi, DOWN *d* = 0.40 for neurofeedback hemi, *d* = 0.35 for opp. hemi, *Figure 3*).

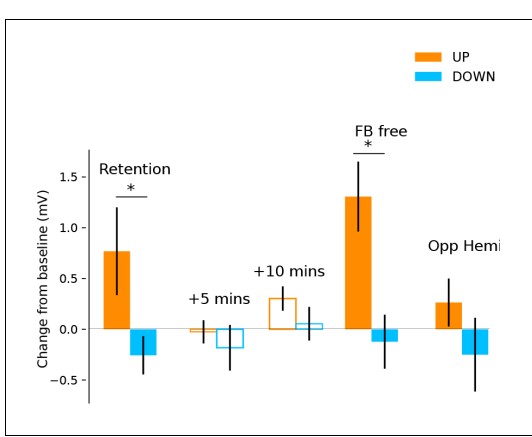

**Figure 3.** Retention, aftereffects and feedback-free measurements. Filled bars represent blocks of neurofeedback, and unfilled bars represent MEPs collected at rest. Shown are MEP amplitudes with their preceding resting baseline values subtracted. Values above 0 represent increases relative to baseline, and below 0 represent decreases. State-dependent neurofeedback training feedback effects were still evident in a retention block carried out approximately 6 months following the initial experiment. No aftereffects were observed on resting MEP amplitude 5 and 10 min later. In a separate block participants were capable of upregulating and downregulating MEP amplitudes with feedback removed (FB free). MEPs measured from the opposite hemisphere during neurofeedback exhibited a similar pattern of modulation.

DOI: https://doi.org/10.7554/eLife.40843.009

## Paired pulse TMS investigation of mechanisms

Finally, we investigated which excitatory/inhibitory circuits may have contributed to the changes in corticomotor excitability, using paired pulse TMS measures of three different neurophysiological processes: (i) Short-interval intracortical inhibition (SICI), believed to reflect postsynaptic GABA$_A$ inhibition (*Fisher et al., 2002*); (ii) Long interval intra-cortical inhibition (LICI), considered as a marker for postsynaptic GABA$_B$ inhibition; and (iii) late-cortical disinhibition (LCD), which is thought to measure presynaptic GABA$_B$ disinhibition, and manifests as a period in which MEP amplitude returns to and typically overshoots baseline levels, in a time window following LICI (~220 ms after a suprathreshold conditioning TMS pulse; *Cash et al., 2011*; *Cash et al., 2010*; *Caux-Dedeystère et al., 2015*). In the following analyses we determined the time point (baseline vs during NF) x neurofeedback type (UP, DOWN) interaction and applied FDR correction for multiple testing. Single pulse MEPs collected during these measurement blocks (25% of all trials) revealed significantly larger MEP amplitudes for the UP than the DOWN condition, replicating the findings of the main experiment (*Figure 4A*; significant time point x neurofeedback type interaction: F(1,27.67) = 14.36, p$_{FDR}$ = 0.001). Surprisingly, there were no significant differences in the magnitude of SICI (% of single pulse MEPs) between the resting baseline data and the SICI MEPs collected during neurofeedback, nor between the UP versus DOWN states (time point x neurofeedback type interaction [F(1,28.31)= 0.08, p$_{FDR}$ = 0.77]). The same was true for LICI (time point x neurofeedback type interaction [F(1,28.90) = 0.02, p$_{FDR}$ = 0.88]). Thus, circuits controlling postsynaptic inhibition did not seem to be differentially modulated by the UP versus DOWN state. However, for LCD there was a significant time point x neurofeedback type interaction (F(1,28.35) = 12.09, p$_{FDR}$ = 0.002, *Figure 4B*).

Pairwise comparisons revealed that LCD was significantly elevated in the UP condition, when compared to the baseline measurement taken immediately before neurofeedback (*Figure 4B*, right panel, MeanDiff = 50.9%, df = 28.35, p < 0.001) and when compared to the equivalent data recorded in the DOWN condition (MeanDiff = 56.1%, df = 28.76, p < 0.001). For the DOWN condition LCD did not differ significantly from baseline (p = 0.45).

## Distinct oscillatory signatures for high versus low corticospinal excitability

As part of the initial training study (see *Figure 2A*, 'Ses 3' for the behavioural results), we investigated whether the two different activity states evoked differential cortical dynamics extracted from EEG recordings which were acquired simultaneously while TMS was being performed to provide neurofeedback of MEP amplitude. As distinct functions have been ascribed to eight different sub-frequency bands across the known range of brain signals (0.1–80 Hz), we now probed whether volitional changes in corticospinal excitability of M1, drives neural activity measured in the delta (0.1–4 Hz), theta (5–7 Hz), low alpha (8–10 Hz), high alpha (11–13 Hz), low beta (14–21 Hz), high beta (22–30 Hz), low gamma (31–50 Hz) and high gamma (51–80 Hz) bands. Using the portion of EEG data collected in the 1.5 s prior to each TMS pulse, we calculated relative power in the UP and DOWN states for the eight frequency bands of interest. At this point, one subject was excluded from further EEG analyses due to extremely high Gamma values exceeding 3xSDs above the mean. *Figure 5a–f* (n = 14) shows that UP- versus DOWN regulating corticomotor excitability caused reduced band-limited power in the theta and alpha band (blue areas in *Figure 5b–d*) while gamma power was clearly increased. (red areas in *Figure 5e,f*). For each participant, we extracted the information for the electrode closest to their individual motor hotspot (*Figure 5a* shows the different locations across participants) and calculated whether the UP-DOWN difference (Δ relative power %) deviated significantly from 0 (*Figure 5g*). Wilcoxon signed rank tests revealed significantly higher power for the UP than DOWN condition in the Delta (p$_{FDR}$ = 0.024, d = 0.754), Low Gamma (p$_{FDR}$ = 0.024, d = 0.753) and High Gamma (p$_{FDR}$ = 0.016, d = 0.712) band and significantly lower power for UP than DOWN in the theta (p$_{FDR}$ = 0.003, d = 0.947), low alpha (p$_{FDR}$ = 0.004, d = 0.805) and high alpha band (p$_{FDR}$ = 0.007, d = 0.714). Although the feedback was lateralised to MEPs from the right limb (left hemisphere motor hotspot), we also quantified the same neural oscillations at the corresponding location in the opposite hemisphere. Here, only the theta rhythm showed significantly lower power for the UP than the DOWN state (p$_{FDR}$ < 0.001, d = 1.071, see *Figure 5—figure supplement 1*). Further, in a subset of five participants from the control group who also underwent EEG recording,

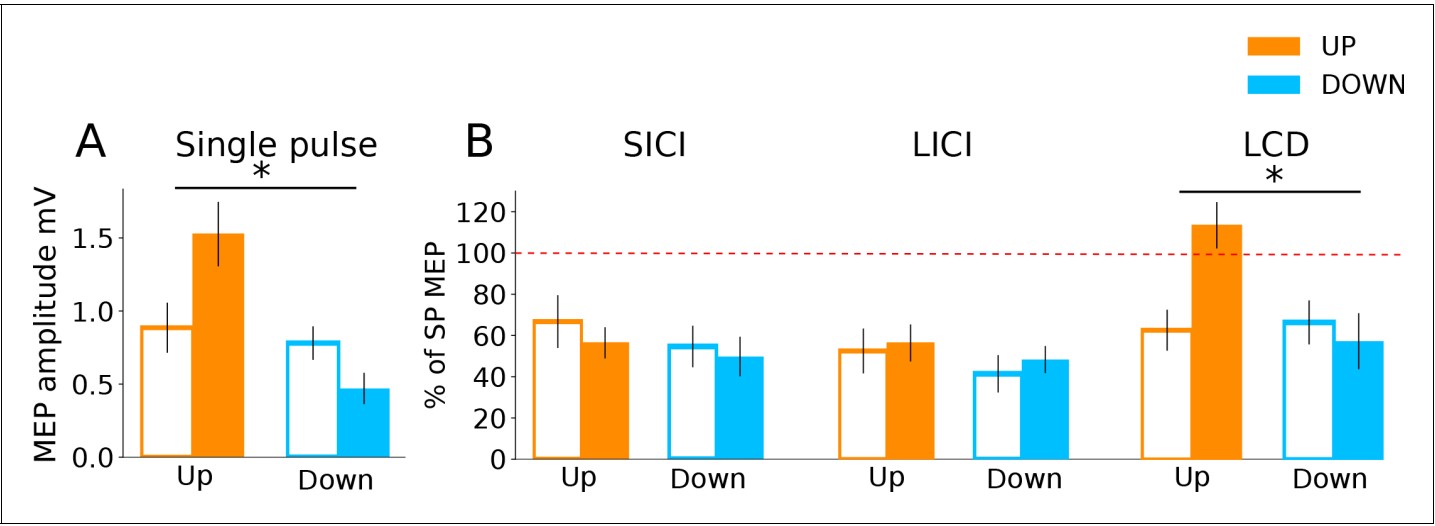

**Figure 4.** Investigation into mechanisms of MEP neurofeedback. The data show paired pulse TMS measurements taken during neurofeedback blocks to probe distinct neurophysiological processes. In all subsequent panels, unfilled bars represent baseline MEP amplitudes collected at rest prior to the block. Panel (**A**) shows that MEP amplitudes from the single pulses (from which neurofeedback was provided) exhibited the same state-dependent modulation as observed previously. In Panel (**B**) MEP amplitudes are expressed as a percentage of the single pulse (SP) MEPs. While expected levels of inhibition were observed for both SICI and LICI paired pulses, there was no state-dependent modulation. LCD was, however, significantly increased in the UP condition relative to baseline, and relative to the DOWN condition.

DOI: https://doi.org/10.7554/eLife.40843.010

more pronounced Beta desynchronization was observed in the UP relative to the DOWN state, but no systematic elevations in low or high Gamma (see *Figure 5—figure supplement 4*).

Next, we tested whether the amplitude of neural oscillations recorded at the hotspot at the time of each TMS pulse could predict the amplitude of the resulting MEPs. For each participant, MEP amplitudes of the 120 trials (60 UP, 60 DOWN) were entered as the outcome variable in a robust regression model with trial-by-trial relative power values for each frequency band as predictor variables. Regression slopes (beta values) for each participant were carried forward into a group level analysis (*Figure 5m*), and Wilcoxon signed rank tests were used to establish whether the slopes were significantly different from 0 (a 0 slope would indicate no statistical relationship between predictor and outcome variable). Lower amplitude oscillations in theta ($p_{FDR}$ = 0.024, *Figure 5h*), low alpha ($p_{FDR}$ < 0.001) and high alpha ($p_{FDR}$ = 0.002) were predictive of larger MEP amplitudes, and higher amplitude oscillations in low gamma ($p_{FDR}$ = 0.020) and high gamma ($p_{FDR}$ = 0.020) were significant predictors of larger MEP amplitudes. In a previous study, it was reported that a strong predictor of cortical excitability was the low gamma: high alpha ratio (*Zarkowski et al., 2006*). We replicated this finding, demonstrating that this ratio was a significant predictor of MEP amplitude ($p_{FDR}$ = 0.016) with larger ratios predicting larger MEP amplitudes.

## EEG data classification

We next tested whether the distinction between the two trained states was large enough that the individual data trials could be successfully predicted as 'UP' state or 'DOWN' state, using machine learning based solely on the EEG power values (relative power data, scaled by 1/f transformation, in the 1.5 s prior to TMS) of the eight frequency bands of interest. A linear support vector machine (SVM) was applied to the data of each participant (60 UP 60 DOWN epochs). The SVM has been shown to be particularly powerful on EEG data, which is noisy and contains many features that are correlated. This approach additionally allowed us to perform feature selection, to quantify which EEG features most heavily contributed to the distinction between the two states. Using only data from the electrode closest to the hotspot (eight rhythms plus LowGamma:HighAlpha ratio) the SVM was capable of classifying the brain states with an average accuracy of 81.5% (±5.1%) based on 10-fold cross validation which differed significantly (p = 0.001, n = 14) from a null model revealed by permutation testing (accuracy null model: 49.0% ± 13). Additionally, incorporating data from the

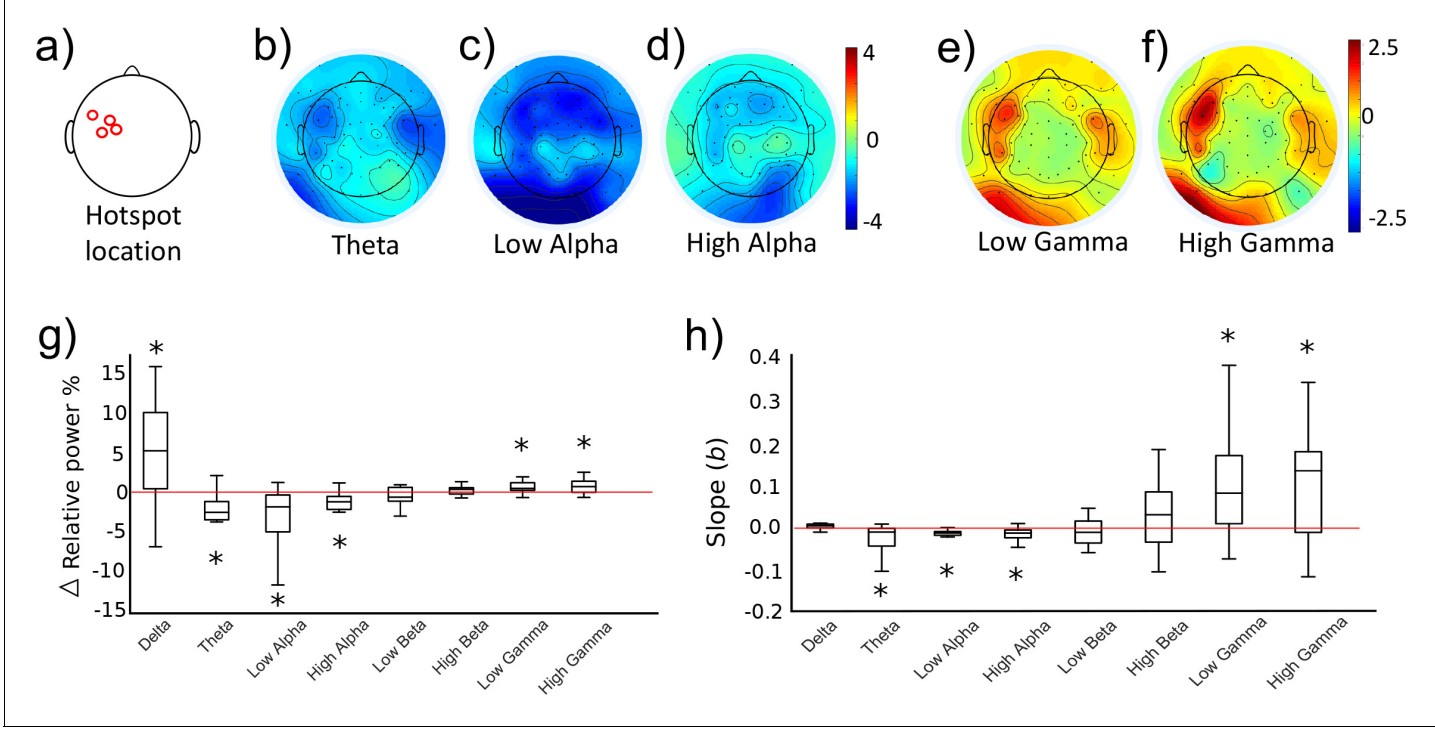

**Figure 5.** Neural oscillations associated with the trained brain states. Panels (**b-f**) show topographical representations of the relative power (in % of whole spectrum) in the UP condition minus the DOWN condition, for five distinct frequency bands (averaged group data, n = 14, three other frequency bands shown in *Figure 5—figure supplement 2*). Red colours indicate regions that demonstrated greater synchronisation in the UP condition. Blue colours indicate greater synchronization in the DOWN condition. The location of the electrode nearest to the TMS hotspot varied between participants but was always within the region indicated in panel (**a**). Colours are scaled from blue-red by minimum-maximum (range shown to right of each plot). Panel (**g**) shows the same data (UP-DOWN) extracted for each participant's hotspot electrode. Values greater than 0 indicate larger amplitude oscillations in the UP condition, and lower than 0 indicate larger oscillations in the DOWN condition. Stars indicate significant deviations from 0 (Wilcoxon Signed Rank tests). Panel (**h**) shows group level data for regression analyses performed on MEP amplitudes with relative power in each frequency band. This included all 120 trials (60 UP, 60 DOWN) collected during the combined TMS-EEG recording session. The Y axis depicts the slope of the regression model. Stars indicate significant deviations from 0 (0 would indicate no slope, Wilcoxon Signed Rank test). Individual regression plots are shown for one representative participant in *Figure 5—figure supplement 3*.

DOI: https://doi.org/10.7554/eLife.40843.011

The following figure supplements are available for figure 5:

**Figure supplement 1.** Boxplots showing relative power in the Up condition minus the Down condition for the electrode corresponding to the hotspot in the opposite (non-feedback) hemisphere.
DOI: https://doi.org/10.7554/eLife.40843.012

**Figure supplement 2.** Non-significant frequency band topographies.
DOI: https://doi.org/10.7554/eLife.40843.013

**Figure supplement 3.** Regression plots of MEP amplitude with relative power for one representative subject.
DOI: https://doi.org/10.7554/eLife.40843.014

**Figure supplement 4.** Control group EEG data.
DOI: https://doi.org/10.7554/eLife.40843.015

same rhythms recorded at the corresponding electrode in the opposite (non-feedback) hemisphere increased this accuracy to 85.1% (±4.6%) across participants (see *Supplementary file 1* Table 1). Using feature ranking based on Recursive Feature Elimination (RFE), taking the mode of the top ranked features across participants revealed that the strongest contribution to the high classification accuracy of the latter SVM was the High Gamma rhythm in the hotspot electrode, followed by High Alpha at the hotspot, then the LowGamma:HighAlpha ratio (for full ranking order see *Supplementary file 1* Table 1).

# Discussion

Here we aimed to uncover neural activity evoked by voluntarily facilitating or suppressing excitability within sensorimotor circuits, while keeping motor or muscle contraction-related sensory feedback constant. We show that using a bidirectional TMS-neurofeedback approach is critical to gain volitional control over MEP amplitudes, a skill that is retained for at least 6 months without further training. This voluntary state-setting with a large dynamic range was related to variation in a paired-pulse TMS proxy measure of pre-synaptic $GABA_B$ mediated disinhibition, and to a prominent increase of gamma power in sensorimotor cortex for the UP state which was accompanied by a clear reduction of power in the theta, low and high alpha bands.

Previous studies have shown that it is possible to gain voluntary control over activity in the central nervous system if appropriate neurofeedback is embedded in a reinforcement learning task, with food rewards for animals (*Fetz, 2013*; *Engelhard et al., 2013*) and visually rewarding stimuli for humans (*Thompson et al., 2009*; *Majid et al., 2015*). Here we confirm that this approach is also suitable for learning how corticomotor excitability can be bidirectionally up- or down-regulated. Our participants were initially familiarized with two motor imagery strategies which are known to modulate corticospinal excitability in the required manner (*Majid et al., 2015*; *Stinear et al., 2006*; *Izumi et al., 1995*; *Fadiga et al., 1999*). Learning, however, indicated by progressively stronger dissociation between the UP and the DOWN state, only took place when direct low-latency feedback regarding the MEP amplitude was provided. After training, participants were able to modulate corticomotor excitability across a large range so that MEP amplitudes were approximately twice as large during the UP than the DOWN condition. UP training, in particular, resulted in an 83.8% increase of MEP amplitudes from baseline, while downregulation of MEP amplitude was possible (e. g. see *Majid et al., 2015*) but more difficult (30.6% decrease from baseline). Once acquired, volitional control of corticomotor excitability was retained for at least 6 months and could be performed even without online feedback indicating true, long-term learning (*Magill, 2010*).

Once participants could control their corticomotor excitability, we uncovered the electrophysiological underpinnings by applying measurements that were independent of the feedback modality (single pulse TMS) and investigated whether there were differences between the UP versus DOWN state. This approach cancelled out the effects that were common to both mental strategies, isolating the mechanisms underlying the MEP modulation. This revealed two key novel electrophysiological findings, involving presynaptic $GABA_B$ disinhibition, and gamma oscillations. Additionally, the pronounced changes of cortical physiology despite the absence of motor or muscle contraction-related sensory input suggests that the increase vs decrease of corticomotor excitability was -at least partly-of cortical origin rather than mediated by spinal cord mechanisms.

The UP state was associated with a significant increase of LCD while other measurements probing inhibitory M1 circuits failed to reveal differential effects for the UP versus DOWN state. LCD is thought to represent a read-out of the presynaptic self-inhibition of GABAergic neurons which is thought to be mediated by extrasynaptic $GABA_B$ auto-receptors (*Mott and Lewis, 1991*; *Sanger et al., 2001*). This mechanism is hypothesized to result in a net facilitatory effect as observed during the UP condition in our study. Previously, LCD was found to be elevated during motor imagery (MI), but this increase relative to rest was observed irrespective of whether participants imagined voluntarily activating or relaxing hand muscles (*Chong and Stinear, 2017*). However, this investigation was conducted in a single session, and did not employ neurofeedback, so MEP modulation by these two imagination conditions could be expected to be substantially smaller than observed in our study, particularly for the voluntary relaxation condition which had a similar excitability state as the rest condition. Thus, it is possible that the clear modulation of LCD observed here only manifested after neurofeedback training, that is when the two excitability states became clearly distinct. It is important to note here that in our protocol, LCD was detected only during the UP state and not at rest. In our search procedure to decide upon the optimum conditioning stimulus (CS) intensities, we prioritized SICI and LICI, finding a CS intensity that elicited as close to 50% inhibition of the test MEP as possible. We tested intensities between 106–114% RMT for LICI (and above or below this if no appropriate inhibition was found), and applied these also to LCD (such that the only difference between the LCD and LICI protocols was the ISI). This may have simply been too low to consistently elicit LCD, which appears to be more robustly evoked at higher intensities (*Cash et al., 2010*; *Cash et al., 2016*) or during mild contraction (*Caux-Dedeystère et al., 2014*), and is not observed

in all individuals (*Cash et al., 2016*). It may also be that LCD is more readily observed using triple pulse procedures in which disinhibition can more directly be measured by the reduction of SICI following a priming stimulus (*Sanger et al., 2001*; *Cash et al., 2016*; *Ni et al., 2007*). The increase in MEP amplitude observed in the TMS-feedback induced UP state, may also be related to the increase in I-wave recruitment that has been reported during LCD (*Cash et al., 2011*) and suggests that further research into this effect is warranted, with parameters more specifically tailored to investigate LCD.

The potential contribution of I-wave mechanisms to the current results can be speculated upon by considering the literature relating disinhibition to short-interval intracortical facilitation (SICF). SICF is mediated by excitatory networks in the motor cortex, which are believed to be responsible for the production of I-waves at a periodicity of around 1.5 ms. When measured in the presence of inhibition (eg. during the LICI time period), the magnitude of SICF remains unchanged, whereas if measured during the period of disinhibition (LCD), SICF is found to be enhanced (*Cash et al., 2011*). In the study by *Cash et al. (2011)*, it was concluded that disinhibition, rather than inhibition, modulates the excitatory networks of SICF, and therefore of I-wave generation. They further suggest that the increase in SICF in periods of disinhibition supports the view that this is an opportune time period to achieve optimal modulation of synaptic networks while they are in a state that is less constrained by ongoing inhibitory processes. In the context of the current results, it may accordingly be predicted that SICF (and recruitment of I-waves in general) would be increased in magnitude during the UP condition, but this is a speculation that will require further experimental testing.

We observed significant modulation of the alpha and gamma rhythms close to M1 of the trained hemisphere. Focusing on data from the recording electrode closest to each individual's hotspot revealed a significant association between low alpha and high gamma power for the UP versus DOWN state. Trial-by-trial modulation of these rhythms correlated significantly with MEP amplitude, and a support vector machine (SVM) classifying the two states based on EEG data ranked the high gamma and high alpha band as the two top features characterizing the distinction. Our observation of reciprocal changes in the alpha and gamma band are in line with previous studies using transcranial as well as intracranial recording methods (*Jensen and Mazaheri, 2010*). The 'pulsed inhibition' theory suggests that repeated bursts of inhibitory alpha activity serve to temporarily silence gamma oscillations (*Jensen and Mazaheri, 2010*). Thus, these two rhythms are seen to exhibit a reciprocal relationship, whereby when alpha is high, gamma is low. In periods of high alpha, gamma may still burst periodically, but only at the troughs of the oscillation cycle, meaning that the gamma 'duty cycle' (window for neural processing) is short, and only brief messages can be sent. By contrast, in periods of low alpha power, the gamma duty cycle is longer, and more extensive neuronal processing and inter-regional communication may occur. Our finding of increased gamma activity is also consistent with previous animal literature, showing that the pharmacological removal of $GABA_B$-mediated inhibition (by receptor blockage) in rats results in increased gamma oscillations (*Leung and Shen, 2007*) which have been shown to be largest in M1's layer V (*Johnson et al., 2017*).

Gamma has often been considered difficult to detect using scalp electrodes because it is highly localised (*Buzsaki, 2006*) and may also reflect non-cortical sources when recorded with EEG (*Yuval-Greenberg et al., 2008*; *Whitham et al., 2007*). However, it is tempting to speculate that, in our experiment, gamma activity was strongly synchronized as a consequence of the neurofeedback training, where participants learned to substantially facilitate corticomotor excitability while keeping EMG activity constant, such that EMG amplitude differed only minimally between the UP and DOWN conditions. This suggestion is in line with previous neurofeedback studies that provided direct feedback of gamma activity, showing that gamma power could be upregulated to a substantial amount which even exceeded power values observed during movement execution (*Engelhard et al., 2013*; *Miller et al., 2010*). By keeping the visual feedback for the two conditions identical, we ensured that differences in eye movements between the UP and DOWN states were minor. As we were particularly interested in gamma oscillations, we additionally performed all EEG recordings in an electromagnetically shielded room, using a gel-based electrode system to maximize signal to noise ratio.

Previous studies have taken a correlational approach to investigating the relationships between brain rhythms and corticomotor excitability. These have shown that low alpha (*Sauseng et al., 2009*; *Zrenner et al., 2018*) or beta power (*Mäki and Ilmoniemi, 2010*) as well as high gamma power

(*Zarkowski et al., 2006*) during natural fluctuations at rest are associated with larger MEP amplitudes. We confirm and extend these results by introducing causality to this relationship for the first time, showing that experimentally driving excitability into two distinct states causes specific patterns of neural dynamics in the volitionally controlled cortical area.

While changes in alpha and gamma were specific to the hemisphere from which feedback was provided, theta showed a bilateral pattern of modulation, being higher in the DOWN than the UP state in motor areas in both hemispheres. While mid-frontal theta activity has been linked to error monitoring (*Pezzetta et al., 2018*) the role of lateralized theta activity close to the sensorimotor hotspot electrode and its symmetric counterpart is less clear. Slower rhythms exert effects over larger distances, and are thought to be involved in long-range communication (*Buzsaki, 2006*). A similar pattern of upregulation and downregulation was observed in the homologous muscle in the opposite limb, albeit weaker and not statistically significant. This is likely a reflection of the extensive transcallosal structural connectivity and functional coupling of homologous regions of the cortical motor network (*Ruddy et al., 2017a*; *Ruddy et al., 2017b*; *Ruddy, 2017*). It is tempting to speculate that the bilateral theta activity observed in the current study served to regulate the inhibition/facilitation of functional coupling or 'spillover' of activation from motor areas in the target hemisphere to their homologous counterparts.

Surprisingly we did not observe differential modulation of the Beta band, which is the predominant oscillatory frequency in sensorimotor cortical regions (*Baker et al., 1997*; *Murthy and Fetz, 1996*). It typically desynchronizes (together with alpha) during motor execution and motor imagery (*Ramos-Murguialday and Birbaumer, 2015*; *Pfurtscheller and Neuper, 1997*; *Pfurtscheller and Berghold, 1989*; *Alegre et al., 2004*) and has been associated with corticomotor excitability at rest (*Zarkowski et al., 2006*). As our results represent the direct contrast between the UP and DOWN states, the lack of Beta involvement may firstly be due to the fact that both conditions involved a mental strategy targeted at the sensorimotor system and, secondly, that no temporal structure was imposed so that we could not perform analyses which are, for example, time-locked to the potential onset of these mental strategies. However, our data further confirm that the two 'inhibitory' rhythms alpha and beta might serve different functions in selecting and activating the appropriate sensorimotor representations (*Brinkman et al., 2014*).

Here we present an innovative approach to voluntarily and bidirectionally change the state of the motor cortex, by directly targeting MEP amplitudes in a neurofeedback paradigm. This method provided a unique opportunity to reveal the oscillatory and neurochemical underpinnings of the two distinct trained brain states, using concurrent TMS EEG measurements, and mechanistic follow-up investigations using paired-pulse TMS. The results comparing UP and DOWN states indicate that voluntary upregulation of corticomotor excitability causes increased presynaptic $GABA_B$-mediated disinhibition, elevated neural oscillations in the gamma frequency range, and reduced alpha and theta rhythms.

This paves the way for new technologies that allow the user to regulate aspects of their own brain function in order to reach desired states that are, for example, associated with enhanced motor performance. In the context of stroke rehabilitation, training volitional modulation of corticomotor excitability may hold promise as a rehabilitative therapy early after stroke, that is when patients are deprived of rehabilitation training because they are unable to execute overt movements with the impaired upper limb. As it is known that LCD is recruited during actual movement (*Caux-Dedeystère et al., 2015*; *Hammond and Vallence, 2007*; *Opie et al., 2015*), the elevated LCD we observed in the UP condition may reflect that the neurofeedback had engaged similar mechanisms to those involved in movement execution, using only voluntary endogenous processes. Furthermore, as pathological hyperexcitability of the non-damaged hemisphere has been hypothesized to limit recovery in some patients (*Murase et al., 2004*), the TMS-neurofeedback protocol can be individually tailored either to upregulate the damaged hemisphere, down-regulate the intact hemisphere, or a combination of both, depending on the patient's specific needs.

# Materials and methods

## Participants

Fifteen healthy volunteers (age 23 ± 3.14 s.d, seven female) participated in the experimental group. An additional thirteen participants (age 25 ± 3.06 s.d, three female) formed a control group. A power analysis was conducted a-priori to estimate appropriate sample size. This is reported in *Supplementary file 1* table 5. All participants were right handed according to the Edinburgh Handedness Inventory (*Oldfield, 1971*), and gave informed consent to procedures. The experiments were approved by the Kantonale Ethikkommission Zürich (KEK-ZH-Nr. 2014–0242), and were conducted in accordance with the Declaration of Helsinki (*World Medical Association, 2013*).

## TMS-based neurofeedback

Participants undertook five sessions of TMS-based neurofeedback, on separate days. The first four days comprised of neurofeedback training, and on the fifth day neurofeedback was performed with simultaneous electroencephalography (EEG) recording to investigate state specific neural dynamics. On two of the training days neurofeedback was adjusted so that a rewarding visual stimulus was displayed when MEPs were larger than baseline (the 'UP' condition) and on the other two days, the rewarding stimulus was displayed when MEPs were smaller than baseline (the 'DOWN' condition). On each of the training days, four separate blocks of neurofeedback were preformed, each comprising of 30 individual MEP feedback trials (total 120 trials per day). The format of individual trials and feedback is described in more detail below, and an outline of experimental design is shown in *Supplementary file 1* Table 7. Baseline corticospinal excitability was measured on each day prior to training (20 MEPs) and post-measurements were taken during the rest periods between each of the four blocks (12 MEPs per measurement).

Subjects sat in a comfortable chair with both arms and legs resting in a neutral position supported by foam pillows. Surface electromyography (EMG, Trigno Wireless; Delsys) was recorded from right First Dorsal Interosseous (FDI), Abductor Digiti Minimi (ADM), Opponens Pollicis (OP), and left FDI. EMG data were sampled at 2000 Hz (National Instruments, Austin, Texas), amplified and stored on a PC for off-line analysis.

TMS was performed with a figure-of-eight coil (internal coil diameter 50 mm) connected to a Magstim 200 stimulator (Magstim, Whitland, UK). The coil was held on the left hemisphere over the 'hotspot' of the right FDI at the location with the largest and most consistent MEPs, and with the optimal orientation for evoking a descending volley in the corticospinal tract (approximately 45 degrees from the sagittal plane in order to induce posterior-anterior current flow). Once the hotspot was established, the lowest stimulation intensity at which MEPs with peak-to-peak amplitude of approximately 50μV were evoked in at least 5 of 10 consecutive trials was taken as Resting Motor Threshold (RMT).

The stimulation intensity used to evoke MEPs during the experiment was chosen using the following procedure in order to obtain a baseline MEP amplitude that was 50% of the participant's maximum. A recruitment curve *Carson et al. (2013)* was performed at the beginning of the first experimental session, whereby 6 TMS pulses were applied at 10 different intensities relative to RMT (90%, 100%, 110%, 120%, 130%, 140%, 150%, 160%, 180%, 190%) in a randomized order. MEP amplitude at each intensity was plotted to determine the point on the curve at which plateau occurs and the MEPs do not continue to increase. Maximal MEP amplitude was recorded, and the intensity required to evoke 50% of this amplitude was used for all subsequent testing. With this approach, there was scope for MEP amplitude to both increase and decrease to similar extents from this 'intermediate' value. Post-hoc analyses revealed that this procedure resulted in an average stimulation intensity corresponding to 130% RMT. Immediately following this procedure and prior to the first block of neurofeedback, 20 MEPs were collected at rest at the chosen intensity to determine 'baseline' corticospinal excitability. The mean MEP amplitude at baseline was recorded and used during neurofeedback to establish the criterion amplitude that determined whether participants received either positive or negative feedback.

## Format of neurofeedback

Neurofeedback was performed using custom written MATLAB software (*Ruddy, 2018*; copy archived at https://github.com/elifesciences-publications/TMS-neurofeedback). Participants kept eyes open with attention directed to a monitor in front of them. They were instructed to relax their limbs and avoid tensing any muscles throughout the experiment. In order to ensure that MEP amplitude could not be influenced by background muscle activation, the root mean square (rms) of the EMG signal for each muscle for the previous 100 ms of data was calculated and displayed in real-time on screen at the beginning of each trial in the form of four coloured 'traffic lights', representing each muscle (*Figure 1A*). If the background EMG in a muscle exceeded 7μV, the corresponding light immediately turned red, and stayed red until the EMG dropped below 7μV. Even when the rms EMG dropped below 7 μV, this had to be sustained for 500 ms in order for the software to trigger the remainder of the trial to commence. When this criterion was met, the traffic lights were replaced by a fixation cross which appeared in the centre of the screen indicating that the trial has started. After a variable period of time (between 5.5–8.5 s, or longer if the trials was paused due to muscle activation as described below) a TMS pulse was fired. This variable fixation interval was included to ensure that participants were unable to anticipate the pulse timing, or engage a timely EMG-based strategy just prior to TMS delivery. Note that also during this variable fixation interval background EMG continued to be monitored in the same fashion as prior to the trial, that is the trial was automatically paused until rms EMG of all muscles dropped below 7 μV for at least 500 ms.

The MEP amplitude for the target muscle (right FDI) was immediately measured and displayed to the participant on screen within 500 ms. The display consisted of a vertical bar indicating MEP amplitude relative to a horizontal line in the middle of the screen representing the mean recorded at baseline (*Figure 1B*). In 'UP' sessions if the MEP was larger than the criterion amplitude, the bar was shown as green with a tick beside it, a positive sound bite was heard, and a number adjacent to a dollar sign incremented to indicate that a small financial reward had been gained. If the MEP was smaller than the criterion amplitude, the bar was red with a cross beside it, a negative sound bite was heard, and no financial reward was shown. The reverse was true in the 'DOWN' sessions (*Figure 1C*). The feedback remained on screen for 4 s, before being replaced by the traffic lights display preceding the next trial. Participants were instructed to attend to the feedback and that the goal was to increase (or decrease) the size of the MEP represented by the bar. Prior to the experiment participants read an instruction sheet explaining the procedures above and providing recommended mental strategies that were reported in previous literature in which corticospinal excitability was downregulated (*Majid et al., 2015*) and upregulated (*Izumi et al., 1995*) by motor imagery (Specific task instructions are provided in Supplementary Text. Also see *Supplementary file 1* Table 6). Initially the criterion amplitude corresponded to the baseline MEP measure. After each block of 30 MEPs, performance was quantified and the task difficulty was adjusted if necessary. If the success rate was >70% difficulty was increased by raising (or lowering in the DOWN condition) the criterion MEP amplitude that needed to be reached by 10% in order for the positive reward to be presented. If performance was >90%, this was adjusted by 20%.

## EEG session

On the fifth day neurofeedback was provided during simultaneous EEG recording. The participant's TMS hotspot was determined and marked on the scalp prior to EEG capping. EEG signals were recorded using a 64 channel gel-based TMS-compatible cap (Electrical Geodesics Inc., Oregon, USA), and the channel closest to the TMS hotspot was noted. EEG data were amplified and sampled at 1000 hz. In order to minimize artefacts associated with the direct contact of the TMS coil resting on the electrodes of the EEG cap, we designed and 3D-printed a custom plastic 'coil spacer' device (*Ruddy et al., 2018*), which has four wide legs positioned to provide a platform distributing the weight of the TMS coil, so that it hovers over the electrodes without contact (*Figure 1D*). This allowed quality recordings to be obtained even from the channel of interest closest to the participant's 'hotspot'. The participants RMT was established while wearing the EEG cap with TMS coil spacer, and the same % above threshold that was used for all previous sessions was applied for neurofeedback. Impedances were monitored throughout and maintained below 50 kΩ.

Baseline corticospinal excitability was measured in the same fashion as for the first four sessions, followed by two blocks of neurofeedback (UP or DOWN, counterbalanced) with brief (12 MEP) post measurements following each. After the final post measurement, a 15 min rest break was scheduled for the participant. Following this, the procedure was repeated and baseline excitability was measured again, followed by two blocks of either UP or DOWN neurofeedback (whichever was not performed in the first half of the session). At the end of this session participants were debriefed.

## Control group

Participants were blinded as to whether they were allocated to the experimental or control group. The control group experienced identical conditions to the experimental group, with the exception that direct neurofeedback was not provided. The visual feedback bar demonstrating MEP amplitude was always the same height (reaching the 'mean' horizontal line). 'Positive' feedback/rewards were presented in the same proportion as in the experimental group (66% of all trials - calculated upon completion of experimental group), but at a fixed and predicable rate in order to prevent the development of illusory correlations. Participants were instructed to attend to the visual feedback on screen, and that rewards would occur at a fixed rate. Aside from this, they were otherwise given identical instructions as the participants in the experimental group- that is the same recommended mental strategies were provided on control 'UP' and 'DOWN' blocks.

## Data processing and analysis

### MEP data

EMG data from all four hand muscles were band-pass filtered (30–800 Hz), separately for the portion of data containing the 100 ms of 'pre-TMS' background EMG, and for the portion of EMG containing the MEP, in order to prevent smearing of the MEP into the background EMG data chunk. The root mean square (rms) of the background EMG was calculated, and peak-peak MEP amplitude was measured. Trimming (removal) of the maximum and minimum MEP in each block was performed in order to screen out extreme values. MEP amplitude is known to be modulated by EMG background activation (*Hess et al., 1986*; *Devanne et al., 1997*). Therefore, the rms pre-stimulus EMG recordings were used to assess the presence of unwanted background EMG activity in the period 110 to 10 ms preceding the magnetic pulse. MEPs preceded by background EMG higher than 0.01 mV were excluded. For each subject and over all trials we calculated the mean and standard deviation of the background EMG. MEPs that occurred when the background EMG value exceeded 2.5 standard deviations above the mean, and MEPs with a peak-to-peak amplitude which exceeded $Q3 + 1.5 \times (Q3 - Q1)$ were removed from further analysis, with Q1 denoting the first quartile and Q3 the third quartile computed over the whole set of trials for each subject.

Inferential statistics were computed using Mixed Effects Models in SPSS (Version 16.0, SPSS Inc. Chicago, US), as they account for covariances between related data samples in repeated measures designs, and have greater flexibility for modeling effects over time than traditional ANOVA approaches (*Gueorguieva and Krystal, 2004*). Fitting of the mixed effects models employed restricted maximum likelihood estimation (REML) and a compound symmetry covariance matrix. Model fit indices (Akaike Information Criterion and Schwarz Bayesian Criterion) were considered prior to choosing the covariance matrix and model type. Fixed effects were *neurofeedback type* (UP or DOWN) and *block number* (1-10). The influence of each of the fixed effects on the model was estimated using *F* tests. In all models *subject* was designated as a random effect with random intercepts.

The criterion alpha value was set to 0.05 for all inferential tests. In cases where multiple comparisons were performed, *p* values were false discovery rate (FDR) corrected.

### EEG data

Signals from all 64 channels were first epoched to extract only the data on each trial from the 4 s before the TMS pulse. This was to remove the substantial artefacts that arise during the magnetic pulses, prior to conducting any filtering or further processing. These separate chunks of unpolluted data were then concatenated into one continuous epoch, and highpass filtered at 1 Hz, prior to conducting an independent components analysis (ICA). Independent components were visualized and

those containing artefacts arising from eye movements, facial EMG, cardiac signals, bad channels or other non-brain activity related signals were removed.

The cleaned data were average-referenced, and re-epoched into chunks of data containing only the 1.5 s on each trial prior to the TMS pulse (ie. to capture the ongoing oscillatory activity at the instance in which the TMS occurred, while the fixation cross was on screen and the 'traffic lights' had disappeared).

A power spectrum was computed (Welch periodogram/FFT) for each single epoch and the mean power (and relative power) in each of the relevant bandwidths were extracted (delta (0.1–4 Hz), theta (5–7 Hz), low alpha (8–10 Hz), high alpha (11–13 Hz), low beta (14–21 Hz), high beta (22–30 Hz), low gamma (31–50 Hz) and high gamma (51–80 Hz). Relative power was calculated per channel per epoch, with the power value for each individual frequency point (eg. 20 Hz) being expressed as a % of the total power within the spectrum for that epoch. Thus, the values are normalised across individuals so that large variations in absolute power values can not bias results. Power values were computed separately for UP and DOWN trials, and non-parametric Wilcoxon signed rank tests (with FDR correction) were used to compare neural oscillations in these two states.

We also analysed whether trial-by-trial variation of EEG data were associated with trial-by-trial variation of MEP amplitudes. Therefore, relative power in each bandwidth for each epoch was entered into a multiple regression model with MEP amplitudes measured in the muscle from which neurofeedback was provided (right FDI). The beta (slope) values resulting from each regression model for each participant were forwarded into a group-level analysis.

## Classification of distinct brain states

Individual epochs of EEG data (60 UP, 60 DOWN) were classified by a linear support vector machine (SVM, 10-fold cross validation), to test separately for each participant whether the epochs could be successfully predicted as 'UP' state or 'DOWN' state based solely on the power values (scaled by using 1/f transformed relative power) of the four frequency bands of interest. The SVM was chosen as it is known to perform particularly well in BCI settings using EEG data which is noisy and has features that are correlated. In order to validate the results the same procedure was repeated with randomly permuted labels, and this null model was statistically compared to the model with true labels (C = 1). Feature selection was conducted using feature ranking based on Recursive Feature Elimination (*Guyon et al., 2002*).

## Follow-up experiment 6 months later

A sub-set of 11 participants from the experimental group returned approximately 6 months later to participate in a follow-up experiment probing retention and mechanisms underlying the two distinct states. This was conducted over a further 4 days of testing. On one day, retention, aftereffects, and excitability in the opposite 'untrained' hemisphere were tested for the 'UP' condition. On another, neurophysiolocial mechanisms were probed using paired pulse TMS. These two days were repeated for the 'DOWN' condition, and the order of these sessions was counterbalanced. We additionally tested whether trained participants were able to upregulate and downregulate when feedback was temporarily removed.

## Retention testing and aftereffects measurement

After a 6 month break and no top-up training, participants were tested with one block of TMS-neurofeedback (20 MEPs) in order to assess retention of learning. All other procedures were identical to those carried out in the main experiment.

Following this block, 12 MEPs were collected at rest after 5 and 10 min.

## Excitability in the opposite hemisphere

During one block, two TMS coils were used, placed over the right and left motor hotspots (as described previously). This block contained 40 trials, 20 of which were normal TMS neurofeedback trials. The other 20 were trials where TMS was applied to the opposite hemisphere, rather than to the hemisphere that was the target for neurofeedback. No feedback was given in these trials. The presentation of left and right hemisphere TMS pulses was randomized.

## Feedback-free measurements

We additionally tested whether trained participants were able to upregulate and downregulate when feedback was temporarily removed. In this feedback-free block, the timing of trials and participant instructions were identical to normal neurofeedback blocks, but in place of the usual feedback bar showing MEP amplitude, the white fixation cross simply turned red during this period. The onset of trials was still contingent on muscles being completely relaxed, and the traffic lights display still preceded every trial.

## Paired pulse TMS measurements

On separate days (one 'UP' one 'DOWN') from the measurements described above, we performed three additional blocks of TMS neurofeedback (24 trials per block x 3 = 72 total trials), in which just 25% of trials were standard single pulse TMS-neurofeedback trials, with the usual feedback. The remaining trials contained paired pulses in place of the usual single pulse TMS. For all paired pulse measurements, the test stimulus intensity was identical to that which had been chosen for the TMS neurofeedback (ie. that produced MEPs that were 50% of the maximum on the recruitment curve). On 25% of trials Short Interval Intracortical Inhibition (SICI) was measured. This was with a conditioning stimulus intensity that was chosen using a personalized search procedure testing intensities ranging from 50–90% RMT, to achieve as close to 50% inhibition as possible, and an inter-stimulus interval of 1.97 ms (*Peurala et al., 2008*). The reduction in the size of the test MEP is believed to represent postsynaptic GABA$_A$ inhibition (*Fisher et al., 2002*). On 25% of trials Long Interval Intracortical Inhibition (LICI) was measured. This was with two suprathreshold pulses, with the conditioning stimulus intensity chosen using a search procedure between 106–114% RMT, and an inter-stimulus interval of 100 ms (*Cash et al., 2010*). This is believed to reflect postsynaptic GABA$_B$ inhibition (*McDonnell et al., 2006*). On the remaining 25% of trials, Late Cortical Disinhibition (LCD) was tested. This was with the exact same pulse intensities as used for LICI, but with a 220 ms inter-stimulus interval (*Cash et al., 2010*), and is thought to measure presynaptic GABA$_B$ disinhibition (*Cash et al., 2011*; *Cash et al., 2010*; *Caux-Dedeystère et al., 2015*). The order of presentation of paired pulses and single pulses was randomized.

Baseline measurements were taken at rest with each of these three paired-pulse TMS protocols, prior to the beginning of neurofeedback blocks (20 of each type of paired pulse measurement, and 20 single pulse MEPs).

## Data availability statement

Data are openly available on the ETH Library Research Collection with the DOI: https://doi.org/10.3929/ethz-b-000300799.

## Acknowledgements

This work was supported by grant number 320030_175616 from the Swiss National Science Foundation. KLR is supported by a research fellowship from the Irish Research Council, GOIPD/2017/798. We thank Dr. Robin Cash for helpful comments in preparing this manuscript and Marta Stepian for assistance during data collection.

## Additional information

### Funding

| Funder | Grant reference number | Author |
|---|---|---|
| Swiss National Science Foundation | 320030_175616 | Nicole Wenderoth |
| Irish Research Council | GOIPD/2017/798 | Kathy Ruddy |

The funders had no role in study design, data collection and interpretation, or the decision to submit the work for publication.

## Author contributions

Kathy Ruddy, Conceptualization, Data curation, Formal analysis, Funding acquisition, Investigation, Methodology, Writing—original draft, Project administration, Writing—review and editing; Joshua Balsters, Formal analysis, Supervision, Methodology, Writing—original draft, Writing—review and editing; Dante Mantini, Pegah Kassraian-Fard, Formal analysis, Validation, Methodology, Writing—review and editing; Quanying Liu, Formal analysis, Writing—review and editing; Nadja Enz, Data curation, Methodology, Project administration, Writing—review and editing; Ernest Mihelj, Validation, Visualization, Writing—review and editing; Bankim Subhash Chander, Formal analysis, Methodology, Writing—review and editing; Surjo R Soekadar, Resources, Validation, Methodology, Writing—review and editing; Nicole Wenderoth, Conceptualization, Funding acquisition, Methodology, Writing—original draft, Writing—review and editing

## Author ORCIDs

Kathy Ruddy  http://orcid.org/0000-0001-5501-0423
Joshua Balsters  http://orcid.org/0000-0001-9856-6990
Dante Mantini  http://orcid.org/0000-0001-6485-5559
Quanying Liu  http://orcid.org/0000-0002-2501-7656
Pegah Kassraian-Fard  http://orcid.org/0000-0002-6562-7918
Nadja Enz  http://orcid.org/0000-0003-2476-4710
Ernest Mihelj  http://orcid.org/0000-0002-6080-2553
Bankim Subhash Chander  http://orcid.org/0000-0002-3206-8822
Surjo R Soekadar  http://orcid.org/0000-0003-1280-5538
Nicole Wenderoth  http://orcid.org/0000-0002-3246-9386

## Ethics

Human subjects: All participants gave written informed consent to procedures. The experiments were approved by the Kantonale Ethikkommission Zürich, and were conducted in accordance with the Declaration of Helsinki (1964).

## Decision letter and Author response

Decision letter https://doi.org/10.7554/eLife.40843.021
Author response https://doi.org/10.7554/eLife.40843.022

# Additional files

## Supplementary files

• Supplementary file 1. Supplementary Table 1: Results of linear classifier distinguishing between UP and DOWN states based on EEG power. Also shown are accuracies of 'null models' with random label permutation performed on the same data, along with statistical comparison across subjects of true accuracies vs randomly permuted accuracies (Wilcoxon signed rank tests).

Supplementary Table 2: F tests following mixed effects models comparing background EMG across conditions and blocks. Shown is feedback type (UP,DOWN) by block number interactions on each session.

Supplementary Table 3: Percentage of EMG trials retained. Shown is the percentage (of total number of collected MEP trials) that were retained per block and per session for analysis following postprocessing screening of background EMG amplitude.

Supplementary Table 4: Results of mixed effects models on the percentage of retained background EMG trials during neurofeedback. The dependant variable was the percentage of trials that were retained following screening for background EMG crossing the criterion threshold. Four models were performed; One on data during all neurofeedback trials (10 levels of 'trial') modelling changes over the duration of training (across trials) that were modulated by trial type (UP or DOWN), demonstrating no systematic effects of MEP amplitude modulation on the quantity of trials that were retained for analysis. The three remaining models were conducted separately on the data from Session 1, Session two and Session 3, again revealing no significant modulation on any session. Mixed

models employed a compound symmetry covariance matrix, fixed effects of 'trial' and 'trial type', and random effect of 'participant'.

Supplementary Table 5: F tests following mixed effects models comparing background EMG across conditions and blocks for the retention test conducted months following initial training. Shown is feedback type (UP, DOWN) by block number two levels, baseline + retention) interactions for each of the four recorded muscles.

Supplementary Table 6: In debriefing following the experiment, all participants were asked to write down their final strategies for the UP and DOWN conditions, in their own words.

Supplementary Table 7: Outline of experimental timeline for each day of testing. Each block of data collection is shown in a separate cell, with the beginning of testing indicated by red vertical lines. Each row depicts testing carried out on separate days unless indicated otherwise (Session 3). Blocks in which MEP neurofeedback was conducted are shaded in grey. All other measurements were taken at rest. Sessions 1 and 2 (blocks 1-8) were considered as 'training', as auditory and small financial rewards were presented during feedback. All subsequent blocks of neurofeedback (Session three onwards) were conducted post-training, without financial rewards. Auditory feedback was removed only for the EEG session, to avoid auditory-evoked potentials. The order of UP and DOWN feedback session types was counterbalanced across participants for all sessions. Assessment of resting motor threshold (RMT) was carried out at the beginning of every day of testing. A recruitment curve measurement was performed only at the beginning of the first day in order to establish TMS intensity (as a % of RMT) that evoked MEPs of an amplitude that was 50% of the individual's maximum before plateau. This % of RMT was used on every subsequent session to elicit MEPs during neurofeedback and for resting blocks.

DOI: https://doi.org/10.7554/eLife.40843.016

• Transparent reporting form

DOI: https://doi.org/10.7554/eLife.40843.017

### Data availability

Data are openly available on the ETH Library Research Collection with the DOI: https://doi.org/10.3929/ethz-b-000300799. This contains processed EEG data for all subjects (Figure 5), EMG data and MEPs from the main experiment (Figure 2) and follow-up experiments (Figures 3 and 4). Scripts used for execution of the TMS Neurofeedback experiment are available on GitHub (https://github.com/ncmlabeth/TMS-neurofeedback; copy archived at https://github.com/elifesciences-publications/TMS-neurofeedback).

The following dataset was generated:

| Author(s) | Year | Dataset title | Dataset URL | Database and Identifier |
|---|---|---|---|---|
| Ruddy KL, Balsters JH, Mantini D, Liu Q, Kassraian-Fard P, Enz N, Mihelj E, Chander B, Soekadar S, Wenderoth N | 2018 | MEP neurofeedback | https://doi.org/10.3929/ethz-b-000300799 | ETH Library research collection, 10.3929/ethz-b-000300799 |

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
