## [Decision Letter]

Thank you for submitting your article 'Neural activity related to volitional regulation of cortical excitability' for consideration by *eLife*. Your article has been reviewed by three peer reviewers, including Nicole Swann as the Reviewing Editor and Reviewer #1, and the evaluation has been overseen by a Senior Editor. The following individual involved in review of your submission has agreed to reveal their identity: Preeya Khanna (Reviewer #2).

The reviewers have discussed the reviews with one another and the Reviewing Editor has drafted this decision to help you prepare a revised submission.

Summary:

In this manuscript the authors implemented a novel neurofeedback paradigm designed to test voluntary modulation of corticospinal excitability. Feedback was presented on MEP size in a single-pulse TMS paradigm and participants were instructed to up-regulate or down-regulate MEPs in separate sessions. This manipulation yielded bi-directional differences in MEP size. These effects were retained for a subset of their participants 6 months after initial training. Next, to investigate potential neuro-chemical contributions of this effect, pair-pulse TMS was used and showed a role for GABAB-mediated disinhibition for the up-regulation condition. Finally, CNS contributions to these effects were examined by collecting EEG during the neuromodulation – revealing an increase in gamma activity and a decrease in alpha for the up versus down conditions. By implementing a support vector machine the authors were able to show that the EEG could classify the modulation state (up versus down condition), with a high degree of accuracy.

Overall, the reviewers were impressed with this novel approach for implementing neurofeedback, the author's robust results, and the multiple experimental approaches the authors used. A few of our concerns and points that required additional clarification are described below.

Essential revisions:

1) We understand that the authors conducted a RMS analysis of the EMG data to address concerns that baseline EMG difference could be driving the MEP results. However, we have lingering concerns that EMG differences could still be influencing the MEPs in more subtle ways. We request that the authors perform/include the following analyses to address this concern.

a) Please report the percentage of trials that EMG crossed the threshold both as a function of trial and session. This would help clarify that late in the training, when the subjects are experts, they are not utilizing an EMG strategy.

b) Please include a table similar to Supplementary table 2 in Supplementary file 1 for the 'retention data'. In addition, please clarify how a trial was 'paused' if the EMG crossed a threshold. Does the trial re-start once the EMG is corrected, or is the trial terminated? If a threshold is crossed right before the TMS pulse could this trial still influence the MEP?

2) We were wondering if MEP feedback was the critical aspect driving these results or if differences in strategies used by participants may have played a role. We appreciate that the control group was offered the same suggested strategies, but, since they were also told that the feedback was irrelevant we were wondering if they may have ended up using different strategies. Please report strategies used by both groups. Also please include EEG results (similar to Figure 5G) for the control participants (if available) so that the reader can evaluate to what degree feedback/modulation, versus the strategies participants, used might have driven the EEG findings.

3) It would be helpful to add some more details from individual examples to evaluate how the operant conditioning learning took place. We would like the authors to add individual subject points on top of the bar graphs in Figure 2 (or as separate sub-plots). Additionally, if individual subject learning is not already clear from the revised Figure 2, please add individual plots like in Figure 2A from at least one representative experimental and control subject. Finally, please add trial-by-trial plots (within sessions) of MEP amplitude from an example subject to see how fast learning takes place. This is important to verify that effects are not being driven by a few subjects and also to support the author's argument for potential clinical applications.

4) The experimental design is rich, but also complicated. It would be helpful if the authors included a figure or table to illustrate when sessions occurred and what data was collected during each session.

5) We felt the authors' claim that volitional control over MEP amplitudes were causally related to modulating pre-synaptic GABAB mediated disinhibition was perhaps over-stated. Please add an analysis which correlates LCD and MEP amplitude to support this claim, or soften the claim.

---

## [Author Response]

Essential revisions:1) We understand that the authors conducted a RMS analysis of the EMG data to address concerns that baseline EMG difference could be driving the MEP results. However, we have lingering concerns that EMG differences could still be influencing the MEPs in more subtle ways. We request that the authors perform/include the following analyses to address this concern.a) Please report the percentage of trials that EMG crossed the threshold both as a function of trial and session. This would help clarify that late in the training, when the subjects are experts, they are not utilizing an EMG strategy.b) Please include a table similar to Supplementary table 2 in Supplementary file 1 for the 'retention data'. In addition, please clarify how a trial was 'paused' if the EMG crossed a threshold. Does the trial re-start once the EMG is corrected, or is the trial terminated? If a threshold is crossed right before the TMS pulse could this trial still influence the MEP?

As background EMG could seriously confound our results, we have taken a stringent two-phase approach to control for this, both during data collection (pausing the trials) and in our post-processing (rejecting MEPs from further analysis if the EMG crossed the threshold at the moment the TMS pulse was delivered).

In response to point a) we have now included an additional table in the Supplementary file 1 (Supplementary table 3) reporting what percentage of all MEP trials were retained for analysis following post-processing screening for above-threshold root mean squared (RMS) values (and those exceeding 2.5 x Std above the participant’s mean). These are broken down per session and per trial, and further also per trial type (UP,DOWN). This demonstrates no systematic differences across the 10 trials in the number of rejected MEPs (70.25% retained in Trial 1, 73.78% in trial 10). We have also added inferential statistics to support this argument, in the form of mixed effects models (with exactly the same model structure as used for the MEP analysis and other background EMG analyses already included in the manuscript). This revealed no systematic variations in the number of retained MEP trials over time as a function of trial type (fixed effects for ‘trial- 10 levels’ and ‘trial type- 2 levels, UP+DOWN’). This analysis was first performed on all trials together (i.e. Trials 1-10, to measure systematic changes from early – late training) and also repeated within each session. There was no Trial x Trial Type interaction in any of the 4 models (all p>0.16). Results are reported in the newly added Supplementary table 4 in Supplementary file 1, and referred to in the main manuscript:

'Since MEP amplitudes are a compound measure of excitability influenced by multiple neural elements (Carson, Ruddy and McNickle, 2016), including background muscle activity (Hess, Mills and Murray 1986; Devanne, Lavoie and Capaday, 1997), we repeated this analysis using the root mean squared (rms) background muscle activation (EMG) recorded in the 100ms prior to each TMS pulse. […] Further analyses of background EMG revealed no systematic variation in the number of rejected/retained MEP trials over the course of training, or by feedback type (see Supplementary tables 3 and 4 in Supplementary file 1).'

b) As suggested, we have now included a table similar to Supplementary table 2 in Supplementary file 1 showing inferential analyses of background EMG for the retention data (Supplementary file 5). This demonstrates no significant block (baseline,retention) x Trial Type (UP,DOWN) interactions for any of the four recorded muscles (all p >x). Reference to this new table is made in the manuscript:

'Participants […] retained the ability to upregulate and downregulate their MEP amplitude with neurofeedback (Figure 3; significant effect of neurofeedback type (UP vs. DOWN) in a retention block carried out with no top-up training (F(1,10)=6.64, p=0.028). Again, this was in the absence of any modulation in background muscle activation (Supplementary table 5 in Supplementary file 1).'

We have also further elaborated in the manuscript on the details of how the trial was ‘paused’ if EMG exceeded the 7uV threshold prior to each TMS pulse:

'In order to ensure that MEP amplitude could not be influenced by background muscle activation, the root mean square (rms) of the EMG signal for each muscle for the previous 100ms of data was calculated and displayed in real-time on screen at the beginning of each trial in the form of four coloured ‘traffic lights’, representing each muscle (Figure 1A). […] Note that also during this variable fixation interval background EMG continued to be monitored in the same fashion as prior to the trial, that is the trial was automatically paused until rms EMG of all muscles dropped below 7 μV for at least 500ms.'

2) We were wondering if MEP feedback was the critical aspect driving these results or if differences in strategies used by participants may have played a role. We appreciate that the control group was offered the same suggested strategies, but, since they were also told that the feedback was irrelevant we were wondering if they may have ended up using different strategies. Please report strategies used by both groups. Also please include EEG results (similar to Figure 5G) for the control participants (if available) so that the reader can evaluate to what degree feedback/modulation, versus the strategies participants, used might have driven the EEG findings.

We appreciate that differences in mental strategies can lead to varied patterns of brain activity measured using EEG and also on the amplitudes of MEPs evoked by TMS. In the neurofeedback literature, it is known from animal and primate studies that neural activity can be modified by feedback alone, suggesting a large implicit learning component. In humans, it is likely that implicit learning from feedback still plays a large part in the operant learning process, but additionally on top of this, humans develop explicit mental strategies. Other neurofeedback studies (using EEG for example) have allowed participants to freely develop their own unconstrained strategies, but we felt that this would introduce too much variability into the data. Thus, we decided to provide a recommended starting point strategy for the UP and DOWN conditions (provided in subsections 'Participant Instructions (Experimental group)' and 'Participant Instructions (Control Group)'), identical for both experimental and control group. Thus, any differences in the pattern of brain activity that emerged (and MEP modulation) could be attributed to the neurofeedback. We expect that the neurofeedback provided the experimental group with the opportunity to shape their mental strategy, such that it became optimised by the end of the training to selectively enhance or suppress their MEP amplitudes. The control group did not have such an opportunity, so were unable to learn any contingencies between their ongoing mental activity and the MEPs.

We recorded every participant’s strategy for UP and DOWN in their own words during the debrief following the experiment, and have now reported these in Supplementary table 6 in Supplementary file 1, which is referred to in the last paragraph of the subsection 'Format of neurofeedback'. In short, most participants imagined an action requiring index finger activity in the UP condition and a heavy, cold or absent index finger in the DOWN condition. Based on participants’ self-report during debriefing, there was no systematic difference between the strategies of the experimental and the control group.

The reviewers have suggested including EEG results for the control group (if available) similar to that shown in Figure 5G. While we did not formally record EEG data from all our control participants, we did so for the first 5 subjects before making the decision to drop this aspect of the protocol in the interests of time and prioritisation of resources for the experimental group. These data have now been analysed and a new boxplot similar to that in Figure 5G has been included as Figure 5—figure supplement 4. With such a small number of data points it is difficult to draw conclusions from this analysis, but it is evident that these participants of the control group did not consistently show the elevated Gamma in the UP condition which was clearly present in the experimental group, but that they do show a more pronounced Beta desynchronization (which was not consistently observed in the experimental group). This data and figure have now been referred to in the first paragraph of the subsection 'Distinct oscillatory signatures for high versus low corticospinal excitability'.

3) It would be helpful to add some more details from individual examples to evaluate how the operant conditioning learning took place. We would like the authors to add individual subject points on top of the bar graphs in Figure 2 (or as separate sub-plots). Additionally, if individual subject learning is not already clear from the revised Figure 2, please add individual plots like in Figure 2A from at least one representative experimental and control subject. Finally, please add trial-by-trial plots (within sessions) of MEP amplitude from an example subject to see how fast learning takes place. This is important to verify that effects are not being driven by a few subjects and also to support the author's argument for potential clinical applications.

As per the reviewers suggestion, we now show individual subject’s data using the same convention as in Figure 2A and 2B (i.e. individual datapoints are plotted together with the triangle representing the group mean). Presently this has been included as Figure 2—figure supplement 2. Additionally we followed the reviewers alternative suggestion to add individual plots of single subject data, similar to those in 2A because that shows exemplary changes in MEP amplitudes over time. These are now included as Figure 2—figure supplement 3. One representative participant from the experimental group and one from the control group are shown, plotted on the same scale in mV, representing their baseline-corrected change in MEP amplitude during each of the training trials on each session. From this it can be seen that even on an individual subject level, learning can be observed for both the UP and DOWN states as a result of the operant conditioning procedure.

The participants exhibited different rates of learning across the sample, with some showing signs of robust learning within the first session (first 4 blocks) and others showing later learning profiles. The individual subject shown in Figure 2—figure supplement 3 for the experimental group exhibited learning mainly in the second session. As the reviewers have also requested plots of individual MEP trials in order to assess speed of learning, we have provided these for the same participant shown in Figure 2—figure supplement 3 (as evidence of a ‘slow’ learner), and from one additional participant as evidence of a ‘fast’ learner. This now forms Figure 2—figure supplement 4. We have presented this data in 6 different scatterplots. A and B depict ALL 300 MEPs collected for the slow-learning subject (same as Figure 2—figure supplement 3). A line of best fit is shown in red, based upon a robust regression model. From this it is clear that there is a positive slope in the UP condition (gradual increase in MEP amplitude as trial increments from 1-300 across Sessions 1-3) and a negative slope in the DOWN condition (decrease in MEP amplitudes as trial increments). The robust regression revealed that in both cases the slope was statistically significant (UP: slope=0.006, p<0.0001, DOWN: slope= -0.0006, p=0.008), with Spearmans Rho values of 0.37 (p<0.0001) and -0.20 (p=0.003) respectively.

Importantly, we also want to demonstrate that participants learned at different rates. Thus, in panels C-F we show data from only the first training session (120 MEP trials). It is clear in panels C and D (slow learning subject shown in Figure 2—figure supplement 3 and panels A and B) that there is no learning in the first session (in fact, the slope in DOWN was positive). However, panels E-F show the same data for another participant who exhibited a steep learning curve even in the first session (MEP trials 1-120). In both UP and DOWN, the robust regression result was significant (UP: slope=0.023, p<0.006, DOWN: slope= -0.019, p=0.021), with Rho values of 0.385 (p=0.006) and -0.40 (p=0.007) for UP and DOWN respectively. In the interests of full transparency, we show ALL MEP data points in these plots, including those that were excluded following our stringent background EMG rejection criterion (unfilled dots). Importantly however, these rejected data points were not included in any statistical tests, means, or when computing the lines of best fit (as they may be biased by higher than typical background EMG values).

These new figure supplements have now been referred to in the subsection 'Bidirectional changes in corticospinal excitability were observed in the MEP neurofeedback group but not in a control group', and legends containing a summary of the statistics mentioned above are included at the end of the manuscript.

4) The experimental design is rich, but also complicated. It would be helpful if the authors included a figure or table to illustrate when sessions occurred and what data was collected during each session.

A table displaying the experimental design for each of the separate 9 days of testing is now included as Supplementary table 7 in Supplementary file 1, and referred to in the first paragraph of the subsection 'TMS-based neurofeedback'.

5) We felt the authors' claim that volitional control over MEP amplitudes were causally related to modulating pre-synaptic GABAB mediated disinhibition was perhaps over-stated. Please add an analysis which correlates LCD and MEP amplitude to support this claim, or soften the claim.

As the LCD measurement is calculated as (Paired-pulse MEP amplitude/Single pulse MEP amplitude) * 100, it would not be possible to conduct an analysis correlating the level of LCD with MEP amplitudes (as these are integral to the measure). Thus, we have rather opted for the reviewers second option to soften the claim we make in the manuscript. The following adjustment has been made to our original statement in the opening paragraph of the Discussion:

'This voluntary state-setting with a large dynamic range was related to variation in a paired-pulse TMS proxy measure of pre-synaptic GABAB mediated disinhibition, […]'